# Tissue Immune Profile: A Tool to Predict Response to Neoadjuvant Therapy in Triple Negative Breast Cancer

**DOI:** 10.3390/cancers12092648

**Published:** 2020-09-16

**Authors:** Bruna Cerbelli, Simone Scagnoli, Silvia Mezi, Alessandro De Luca, Simona Pisegna, Maria Ida Amabile, Michela Roberto, Lucio Fortunato, Leopoldo Costarelli, Angelina Pernazza, Lidia Strigari, Carlo Della Rocca, Paolo Marchetti, Giulia d’Amati, Andrea Botticelli

**Affiliations:** 1Department of Radiological, Oncological and Pathological Sciences, Sapienza University of Rome, 00161 Rome, Italy; bruna.cerbelli@uniroma1.it (B.C.); silvia.mezi@uniroma1.it (S.M.); simona.pisegna@uniroma1.it (S.P.); giulia.damati@uniroma1.it (G.d.); 2Department of Medical Surgical Sciences and Translational Medicine, Sapienza University of Rome, 00161 Rome, Italy; 3Department of Surgical Sciences, Sapienza University of Rome, 00161 Rome, Italy; alessandro.deluca@uniroma1.it (A.D.L.); mariaida.amabile@uniroma1.it (M.I.A.); 4Department of Clinical and Molecular Medicine, Sapienza University of Rome, 00187 Rome, Italy; michela.roberto@uniroma1.it (M.R.); paolo.marchetti@uniroma1.it (P.M.); andrea.botticelli@uniroma1.it (A.B.); 5Azienda Ospedaliera San Giovanni-Addolorata, 00184 Rome, Italy; lfortunato@hsangiovanni.roma.it (L.F.); lcostarelli@hsangiovanni.roma.it (L.C.); 6Department of Medical-Surgical Sciences and Biotechnologies, Sapienza University of Rome, 00161 Rome, Italy; angelina.pernazza@uniroma1.it (A.P.); carlo.dellarocca@uniroma1.it (C.D.R.); 7Medical Physics Unit, “S. Orsola-Malpighi” Hospital, 40138 Bologna, Italy; lidia.strigari@aosp.bo.it

**Keywords:** triple-negative breast cancer, neoadjuvant chemotherapy, tissue immune profile, pathological complete response, TILs, CD73, PDL1

## Abstract

**Simple Summary:**

Pathological complete response (pCR) after neoadjuvant chemotherapy can predict survival outcomes in patients with early triple negative breast cancer (TNBC). The immune microenvironment can affect response to chemotherapy. We combined several immune-related biomarkers (TILs, PD-L1 and CD73) in a tissue immune profile (TIP) and investigated if can predict pCR better than single biomarkers in TNBC. The association between TIP and pCR could be proposed as a novel link between immune background and response to chemotherapy. As a future perspective, our results could help to select patients eligible for combinations with immunotherapy or for escalating and de-escalating strategies.

**Abstract:**

Pathological complete response (pCR) after neoadjuvant chemotherapy (NACT) can predict better survival outcomes in patients with early triple negative breast cancer (TNBC). Tumor infiltrating lymphocytes (TILs), Programmed Death-Ligand 1 (PD-L1), and Cluster of Differentiation 73 (CD73) are immune-related biomarkers that can be evaluated in the tumor microenvironment. We investigated if the contemporary expression of these biomarkers combined in a tissue immune profile (TIP) can predict pCR better than single biomarkers in TNBC. Tumor infiltrating lymphocytes (TILs), CD73 expression by cancer cells (CC), and PD-L1 expression by immune cells (IC) were evaluated on pre-NACT biopsies. We defined TIP positive (TIP+) as the simultaneous presence of TILS ≥ 50%, PD-L1 ≥ 1%, and CD73 ≤ 40%. To consider the effects of all significant variables on the pCR, multivariate analysis was performed. Akaike information criterion (AIC) and Bayesian information criterion (BIC) were used for model selection. We retrospectively analyzed 60 biopsies from patients with TNBC who received standard NACT. Pathological complete response was achieved in 23 patients (38.0%). Twelve (20.0%) cases resulted to be TIP+. The pCR rate was significantly different between TIP+ (91.7%) and TIP− (25.0%) (*p* < 0.0001). Using a multivariate analysis, TIP was confirmed as an independent predictive factor of pCR (OR 49.7 (6.30–392.4), *p* < 0.0001). Finally, we compared the efficacy of TIP versus each single biomarker in predicting pCR by AIC and BIC. The combined immune profile is more accurate in predicting pCR (AIC 68.3; BIC 74.5) as compared to single biomarkers. The association between TIP+ and pCR can be proposed as a novel link between immune background and response to chemotherapy in TNBC, highlighting the need to consider an immunological patients’ profile rather than single biomarkers.

## 1. Introduction

Triple-negative breast cancer (TNBC), accounting for 15–20% of all breast cancers (BC), is defined by the absence of oestrogen receptor (ER), progesterone receptor (PgR), and human epidermal growth factor receptor 2 (HER2) expression on cancer cells [1]. It is a heterogeneous disease, including cancers with different gene expression and clinical features [2,3]. As compared to other BC subtypes, it has a worse prognosis, an increased risk of visceral metastasis and fewer therapeutic targets [4,5]. Treatment of early and locally advanced TNBC requires a multidisciplinary approach to reach better outcomes [6]. Anthracycline and taxane-based neoadjuvant chemotherapy (NACT) is the most common choice in early and locally advanced TNBC due to the possibility to achieve a pathological complete response (pCR), which can impact on survival outcomes. In fact, patients who achieve a pCR have better disease-free survival (DFS) and overall survival (OS) compared to patients with residual disease (RD) [7]. Several meta-analysis confirmed the prognostic role of pCR, especially in aggressive BC subtypes such as triple-negative and HER2 positive [8,9,10]. The pCR is a primary endpoint in several studies and could be considered as a surrogate of long term survival outcomes [11]. Despite TNBC is more sensitive to chemotherapy compared to other BC subtypes, only about 30% to 50% of patients will achieve a pCR after NACT [12]. Therefore, the identification of new targets and therapeutic strategies is an urgent clinical need. If compared to other BC subtypes, TNBC appears to be more immunogenic [13]. Moreover, accumulating data suggest that the antitumor activity of conventional chemotherapy is partly due to the enhancement of innate and adaptive immune-response through the triggering of immunogenic cell death (ICD) [14]. Pre-treatment immune activation against cancer cells could influence the response to chemotherapy [15]. Tumor infiltrating lymphocytes (TILs) are a prognostic biomarker in different types of cancer [16]. In TNBC, high TILs have been associated with better prognosis and higher chance to achieve a pCR after NACT [17,18].

The interaction between Programmed cell Death protein 1 (PD-1) and its ligand Programmed Death-Ligand 1 (PD-L1) is one of the main mechanisms of immune escaping and the PD-1/PD-L1 axis represents a therapeutic target in several malignancies [19]. PD-1 is a cell surface protein receptor expressed on both CD8+ cytotoxic and CD4+ helper T cells, B cells, natural killer (NK) cells, dendritic cells, and activated monocytes, while his ligand PD-L1 can be expressed on both immune and cancer cells [20]. After engagement by its ligands, PD-1 indices exhausted T CD8+ and CD4+ lymphocytes and promotes T-regulatory cell proliferation and migration, allowing the immune response to be switched off [21]. In BC, the prognostic role of PD-L1 is still uncertain, with limited and contrasting data [22].

Recently, CD73 has been identified as a further molecular immunosuppressive element in TNBC [23]. CD73 is an ectonucleotidase, expressed on the surface of cancer, stromal, and immune cells that converts adenosine monophosphate (AMP) to adenosine, a soluble immunosuppressive factor. By increasing extracellular adenosine, CD73 suppresses immune response through activation of high-affinity A2A and A2B adenosine receptors. Several types of human cancers overexpress CD73, which has been associated with a poor prognosis [24]. In TNBC, CD73 overexpression seems to be associated with resistance to anthracycline-based chemotherapy and poor prognosis [25,26].

TILs, PD-L1, and CD73 are immune-related biomarkers that can be evaluated in neoplastic cells and tumor microenvironment. Previous studies from our group have shown that both TILs and CD73 can individually predict the response to NACT in TNBC B [27,28]. We now investigate if the contemporary expression of these biomarkers, along with PD-L1, combined in a tissue immune profile (TIP), can be more efficient at identifying patients prone to achieve a pCR.

## 2. Results

Clinical and pathological features of the 60 early TNBC patients in the study population are reported in Table 1.

Median age was 49 (range 28–74). Fifty-nine patients (98.3%) had no special type ductal carcinoma, and 57 (95.0%) had a poorly differentiated (G3) carcinoma. Clinical stage at diagnosis was cT2 in the majority of patients (75.4%) with nodal involvement (cN+) in 46.7% of cases. Thirty-six (60%) patients had a clinical TNM II stage and 24 (40%) III stage. The percentage of Ki67 positive cells was ≥50% in 47 patients (78.4%). A pCR was achieved in 23 patients (38.0%). Thirty-one patients underwent conservative surgery while the remaining received mastectomy. Clinical node status was the only feature significantly associated with response to NACT (*p* = 0.047, Table 2).

The results of tissue biomarkers evaluation on the 60 pre-NACT biopsy specimens are reported in Table 3.

Briefly, TILs were present in all pre-NACT biopsies with values ≥ 50% (H-TILs) in 17 samples (28.4%). The median expression value of CD73 on CC was 40%. Absence of CD73 immunostaining was recorded in seven out of 60 cases (11.6%). Twenty-nine biopsies (48.4%) were classified as high-CD73 with a mean expression score of 63.8% while in the remaining low CD73 group (51.6%) the mean expression value was 15.3%. Interestingly, the median value of CD73 was significantly lower in the H-TILs group compared to the L-TILs one (*p* = 0.01). A positive PD-L1 staining on IC was recorded in 49 out of 60 pre-NACT biopsies (81.7%). The relation between each biomarker and the response to NACT is reported in Table 4.

As expected, H-TILs on pre-NACT biopsies were associated with significantly higher rate of pCR (76.5% vs. 21.5%, *p* = 0.001). Among PD-L1-positive patients, 18 (36.7%) had a pCR and 31 had a RD (63.3%). No significant difference in response to NACT was recorded between the two groups of patients (*p* = 0.734). Patients with PD-L1 ≥ 1% had higher TILs compared to PD-L1 negative (*p* = 0.02, Figure 1). Finally, absent/low CD73 on neoplastic cells was significantly associated with pCR (54.8% vs. 20.6%, *p* = 0.009).

All 60 patients were evaluable for TIP analysis. According to the established criteria, 12 (20.0%) were considered TIP positive (TIP+) and 48 (80.0%) TIP negative (TIP−). A pCR was achieved in eleven (91.7%) out of 12 TIP+ patients compared to 12 (25.0%) out of 48 TIP−. Using a univariate analysis, a positive TIP was significantly associated with pCR (*p* < 0.0001, Table 4). Using a logistic regression for age, tumor proliferation index evaluated by Ki67 and clinical lymph nodes involvement, TIP was confirmed as an independent predictive factor of pCR (OR 49.7 (6.30–392.4)), *p* < 0.0001, Table 5).

Pre-NACT clinical nodal stage was also confirmed as predictive factor (OR 0.11 (0.02–0.62), *p* = 0.013). Finally, we compared the efficacy of TIP versus each single biomarker in predicting pCR by Akaike information criterion (AIC) and Bayesian information criterion (BIC). According to our analysis, the combined immune profile is more accurate in predicting pCR (AIC 68.3; BIC 74.5; Table 6).

## 3. Discussion

Patients who achieve a pCR after NACT have a better prognosis compared to those with residual disease. However, only 30–40% of patients will achieve a pCR after standard treatments. Our study is in line with these data, reporting a pCR rate of 38%. In view of that, the identification of responder or resistant patients is an urgent clinical need. Moreover, novel biomarkers are envisaged, in light of the low predicting value of the clinical and pathological features currently available. In our study population, only cN status was an independent predictive factor of response to NACT. In particular, patients with clinical involvement of axillary lymph nodes had a significant lower pCR rate. Clinical TNM staging was not associated to pCR in our population. This result could be influenced by the relative small sample size; however, it is in line with several previous papers, highlighting the role of cancer biology over extension [9,29,30]. A promising field of research is represented by the study of tumor microenvironment and immune background. Features of pre-existing immune activation in neoplastic tissue, such as TILs ≥50% (high TILs) have been associated with the highest probability to achieve a pCR in TNBC, which is the most immunogenic of breast cancer subtypes [17]. The postulated mechanism could be an increased reaction to drug-mediated immunogenic death by a “ready-to-act” immune environment. A potential biomarker currently under active investigation is CD73, marker of immune suppression [31]. While several studies confirmed the role of CD73 as a prognostic factor, its predictive role is still to be clarified [26]. We previously reported a significant association between low level of CD73 expression by CC (below or equal to the median value in our population, 40%) and pCR in TNBC. A robust marker of pre-existing immune-activation against cancer cells is represented by PD-L1. In TNBC, PD-L1 expression by immune cells seems to be associated with response to various immunotherapy regimens [32]. The cut-off of PD-L1 expression ≥1% on IC is currently used to define PD-L1 positivity in clinical practice [33].

In light of current evidence suggesting a relevant role of the immune microenvironment in the response to NACT we tested the predictive role of TILs, CD73 expression on CC and PD-L1 staining of IC in a sample of TNBC. According to our results, high TILs are strongly associated with pCR (Table 4), in line with previous reports. Repeated statistical analysis on the present study sample confirmed a significant association between absent/low CD73 expression and pCR (Table 4). We found that samples with low TILs had a significantly higher expression of CD73 compared to those with high TILs (*p* = 0.02), as a possible consequence of an immunosuppressive microenvironment.

No significant association between PD-L1 and response to NACT was observed (Table 4).

The landscape of biomarkers is characterized by a plethora of single factors used to predict the response to treatment or the toxicity of therapeutic regimens. However, the idea that a single biomarker could represent the entire dynamic and complex relationship between cancer (metabolism, immunity, genomic profile) and host (clinical features, pharmacogenomics, microbiome, immune system) seems to be too simplistic. In this context, the combination of different biomarkers could be a more promising approach.

According to the main purpose of the study, we have identified a “favorable” tissue immune profile (TIP+) characterized by the simultaneous presence of three features: TILs ≥ 50%, CD73 ≤ 40% on CC, PD-L1 ≥ 1% on IC (Figure 2). We then tested if the tissue immune profile could be more efficient than the single biomarkers at identifying patients more prone to achieve a pCR. In twelve patients (20.0% of the entire population), the pre-NACT samples met the inclusion criteria for TIP+. Eleven (91.7%) out of these 12 patients achieved a pCR. The difference of pCR rate between TIP+ and TIP− was statistically significant (*p* < 0.0001, Table 4). A multivariate analysis including clinical and pathological data confirmed the significant association between TIP+ and pCR (*p* < 0.0001, Table 5). Thus, a positive tissue immune profile is more efficient than each single biomarker in predicting a pCR (Table 6).

This study has some limitations. Firstly, it is a retrospective, single-institute trial. The relatively small sample size of the population studied could be related to unexpected results, such as the borderline correlation that emerged between pCR and disease extension or the lack of association between PD-L1 and pCR. A prospective trial with a larger sample size is necessary to validate our results. Moreover, the benefit obtained using the profile over TILs alone (Table 6) is significant but limited, emphasizing the need for validation on a larger sample. Finally, further evaluation will be needed to confirm the predictive value of TIP after addition of immunotherapy to chemotherapy backbone.

Recent clinical trials have shown promising activity of immune-checkpoint inhibitors on early TNBC, highlighting the need for new immune-related biomarkers. In Keynote 173 trial, the addition of pembrolizumab, an anti PD-1 agent, to standard chemotherapy produced a significant increase of pCR rate in an unselected population [34]. In the phase III GEPAR NUEVO trial, the addition of durvalumab, an another anti PD-1 agent, resulted in an increased rate of pCR from 44% to 53% [35]. In contrast, preliminary results from NeoTRIP trial show no significant difference in pCR adding atezolizumab to a backbone of chemotherapy with weekly carboplatin plus nab-paclitaxel in the same setting [36]. Finally, the results from phase III Keynote-522 trial show that adding pembrolizumab to standard neoadjuvant chemotherapy increased the pCR rate and improve DFS, regardless of PD-L1 expression [37]. Based on these results, it is expected that anti PD-1 agents will soon be introduced in neoadjuvant treatment of early TNBC. As a future perspective, an immune-profile, such as TIP, could improve the selection of patients who could benefit from combination strategies with anti PD-1 agents, considering the high cost and additional toxicities related to therapy escalation.

## 4. Materials and Methods

The clinical-pathological features of the study population have been previously reported [28]. Briefly, based on the availability of adequate biopsy samples, 60 cases were selected from a total of 75 patients with early or locally advanced TNBC consecutively enrolled for neoadjuvant therapy between June 2011 and June 2017. All patients underwent sequential NACT with a standard dose of anthracycline and cyclophosphamide for four cycles every 21 days followed by 12 cycles of weekly paclitaxel or four cycles every three weeks of docetaxel. After NACT, patients underwent surgery (mastectomy or breast conservation with either sentinel lymph node biopsy or axillary dissection as needed). The pathological response was evaluated on post-NACT surgical samples. A pCR was defined as the absence of residual invasive disease at the breast and nodal level (ypT0/is ypN0) [38]. Clinical information, including age and clinical stage, was extracted an anonymized from the institutional databases. The study was conducted in accordance with the Helsinki Declaration and the protocol approved by the institutional ethics committee of the University “La Sapienza” (EC 4181, 25 February 2019).

### 4.1. Tissue Biomarkers Evaluation

Two independent and experienced pathologists evaluated stromal TILs, CD73 expression by cancer cells (CC), and PD-L1 expression by immune cells (IC) in all 60 biopsies. The inter-observer agreement was assessed with the interclass correlation coefficient (ICC). The evaluation of all three biomarkers was necessary to include a sample in our study.

### 4.2. TILs

Hematoxylin-eosin stained slides were evaluated for the presence and percentage of TILs, based on the standardized method proposed by the International TILs Working Group 2014 [39,40]. The result of TILs quantification was reported in terms of 10% increments, from absence (0%) to massive infiltration (90%). Cases with TILs either absent or <50% were classified as low TILs (L-TILs). High TILs (H-TILs) were defined by the presence of TILs ≥ 50%, according to international guidelines [39].

### 4.3. CD73 Expression

The results of CD73 immunohistochemical expression by CC are reported in our previous work. Briefly, two groups of patients were identified: CD73 low, if the percentage of CC expressing CD73 was below or equal to the cut-off of 40% (representing the median value in our sample) and CD73 high if the percentage was above this threshold [28].

### 4.4. PD-L1 Expression

PD-L1 immunostaining was performed with an automated stainer (Benchmark XT, Ventana Medical System, Tucson, AZ, USA) using the SP142 antibody as previously described [27]. PD-L1 expression was evaluated on IC in all the study samples. The threshold of positivity for PD-L1 was set as the presence of immunostaining in ≥1% of IC as previously reported according to the recent approval of atezolizumab in metastatic TNBC [33,41,42].

### 4.5. Combination of Predictive Biomarkers: Tissue Immune Profile (TIP)

To better identify patients more prone to achieve a pCR, we combined together the results of the assessment of the above features to construct a tissue immune profile (TIP). A patient was considered TIP positive (TIP+) based on the combination of TILS ≥ 50%, PD-L1 ≥ 1% on IC and CD73 ≤ 40% on CC in the pre-NACT biopsy (Figure 2).

### 4.6. Statistical Analysis

In the descriptive analysis, the quantitative variables were described as medians and ranges, while qualitative variables were reported as numbers and percentages. Univariate associations between clinical and pathological characteristics and pCR were assessed using the Chi-Squared or Fisher’s exact test, when possible. T-tests for unpaired data were used to compare the medians. The interaction between the clinical and pathological parameters was initially investigated using univariate logistic regression. To consider the effects of all significant variables on the pCR, multivariate logistic regression analysis was performed. Akaike information criterion (AIC) and Bayesian information criterion (BIC) were used for model selection. The optimal model is selected based on the minimum AIC and BIC. The following equations were used to estimate the AIC and BIC of model: AIC = −2 × ln(L) + 2 × k and BIC = −2 × ln(L) + 2 × ln(N) × k, where L is the value of the likelihood, N is the number of patients, and k is the number of estimated parameters [43,44]. The statistical significance was set at *p* < 0.05. SPSS, Version 24 statistical software was used (SPSS Inc. Chicago, IL, USA).

## 5. Conclusions

We provide evidence that a favorable tissue immune profile could be an independent predictive biomarker of pCR, highlighting the need to consider the complexity of tumoral immune response rather than the expression of single biomarkers. However, this observation needs to be confirmed by the analysis of a larger population study.

In conclusion, the association between TIP+ and pCR could be proposed as a novel link between immune background and response to chemotherapy. As a future perspective, our results could help to select patients eligible for combinations with immunotherapy or for escalating and de-escalating strategies.

## Figures and Tables

**Figure 1 cancers-12-02648-f001:**
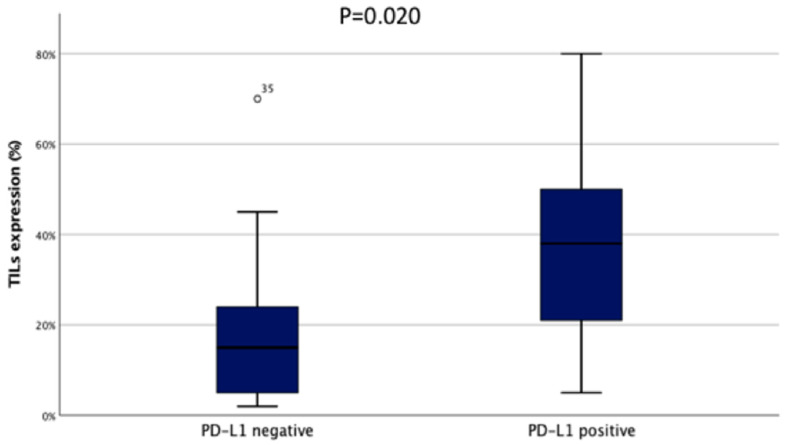
Association between median TILs and PD-L1 status. Figure 1 shows a significantly higher percentage of TILs in PD-L1 positive patients (*p* = 0.02). TILs: tumor-infiltrating lymphocytes.

**Figure 2 cancers-12-02648-f002:**
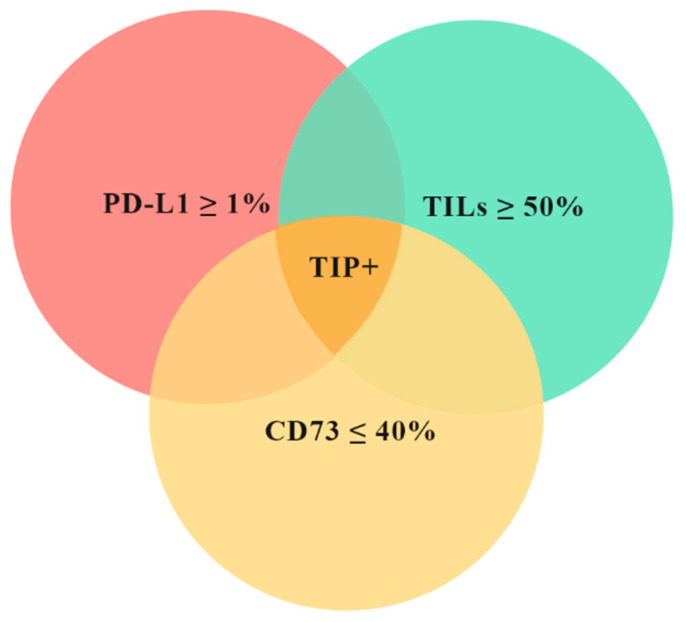
Graphic representation of tissue immune profile (TIP). Figure 2 shows the conceptual Venn diagram of TIP+ patients. The red circle represents patients with PD-L1 ≥ 1%, the green circle patients with TILs ≥ 50% and the yellow circle patients with CD73 ≤ 40%. TIP+ patients are identified by the dark yellow area. TIP+: tissue immune profile positive; TILs: tumor infiltrating lymphocytes.

**Table 1 cancers-12-02648-t001:** Clinical and pathological features of the study population.

Features	N (%)
**Histology**	
Ductal	59 (98.3)
Other	1 (1.7)
**Grade**	
2	3 (5)
3	57 (95)
**Clinical T stage**	
cT1	9 (14.8)
cT2	45 (75.4)
cT3	3 (4.9)
cT4	3 (4.9)
**Clinical N stage**	
cN0	32 (53.3)
cN+	28 (46.7)
**Clinical TNM stage**	
cIIA/IIB	36 (60.0)
cIIIA/IIIB	24 (40.0)
**Ki-67%**	
<50%	13 (21.6)
≥50%	47 (78.4)
**Breast Surgery**	
Conservative	31 (51.6)
Mastectomy	29 (47.5)

Table 1 shows clinical and pathological features of the study population. cN+: node-positive patients; cN0: node-negative patients. Different features are highlighted in bold.

**Table 2 cancers-12-02648-t002:** Association between clinical and pathological features and response to neoadjuvant chemotherapy (NACT).

Features	pCR (%)	RD (%)	*p*-Value
Age			
Median (range)	50 (34–74)	49 (28–73)	0.598
Histology			
Ductal	23 (39.0)	36 (61.0)	
Other	0 (0.0)	1 (100)	0.535
Grade			
2	1 (33.3)	2 (66.7)	
3	22 (37.9)	35 (62.1)	0.684
Clinical T stage			
cT1	5 (55.6)	4 (44.4)	
cT2	17 (37.7)	28 (62.3)	
cT3	0 (0.0)	3 (100)	
cT4	1 (33.3)	2 (66.7)	0.381
Clinical N stage			
cN0	16 (50.0)	16 (50.0)	
cN+	7 (25.0)	21 (75.0)	**0.047**
Clinical TNM stage (II–III)			
IIA/IIB	21 (58.3)	15 (41.7)	
IIIA/IIIB	17 (70.8)	7 (29.2)	0.325
Breast Surgery			
Conservative	10 (35.5)	20 (64.5)	
Mastectomy	12 (41.4)	17 (58.6)	0.321

Table 2 shows the relationship between clinical-pathological features and response to NACT. Age is expressed as median (range) in the pCR group and RD group of patients. pCR: pathological complete response; RD: residual disease; cN+: node-positive patients; cN0: node-negative patients; *p* < 0.05 in bold.

**Table 3 cancers-12-02648-t003:** Pre-NACT biomarkers expression.

Biomarkers	N (%)
TILs	60
≤50%	43 (71.6%)
≥50%	17 (28.4%)
CD73 expression on CC	60
≤40%	31 (51.6%)
>40%	29 (48.4%)
PD-L1 expression on IC	60
≥1%	49 (81.7%)
0	11 (18.3%)

Table 3 shows the different expression of immune-related biomarkers evaluated on pre-NACT biopsies. TILs: tumor infiltrating lymphocytes; CC: cancer cells; IC: immune cells; NACT: neoadjuvant chemotherapy.

**Table 4 cancers-12-02648-t004:** Expression of single biomarkers, tissue immune profile, and response to NACT.

Biomarkers	pCR (%)	RD (%)	*p*-Value
TILs			
Absent/low (<50%)	9 (21.5)	33 (78.5)	
High (≥50%)	13 (76.5)	4 (23.5)	**0.001**
PD-L1 expression on IC			
Positive (≥1%)	18 (36.7)	31 (63.3)	
Negative (0)	5 (44.5)	6 (55.5)	0.734
CD 73 expression on CC			
Median (range)	20 (0–70)	55 (0–100)	**0.008**
Absent/low (≤40%)	17 (54.8)	14 (45.2)	
High (>40%)	6 (20.6)	23 (79.3)	**0.009**
TIP			
positive	11 (91.7)	1 (8.3)	
negative	12 (25)	36 (75)	**<0.0001**

Table 4 shows the association of single biomarkers (TILs, PD-L1, and CD 73) and TIP with response to NACT declined as pCR or residual disease (RD). NACT: neoadjuvant chemotherapy; TILs: tumor infiltrating lymphocytes; TIP: tissue immune profile; CC: cancer cells; IC: immune cells; pCR: pathological complete response; bold values denote statistical significance at the *p* < 0.05 level.

**Table 5 cancers-12-02648-t005:** Association between age, Ki67, cT, cN, TIP, and pCR.

Parameters	OR (95% CI)	*p*-Value
Age (>49 vs. ≤49)	1.67 (0.38–7.27)	0.492
Ki67 (≥50 vs. <50)	2.88 (0.37–22.67)	0.313
cT (cT1/2 vs. cT3/4)	0.39 (0.01–14.92)	0.618
cN (positive vs. negative)	0.15 (0.02–0.92)	**0.041**
TIP (positive vs. negative)	49.7 (6.30–392.4)	**<0.001**

The odds ratio of achieve a pCR using logistic regression analysis. OR: odds ratio; CI: confidence interval; cT: clinical tumor status; cN: clinical node status; TIP: tissue immune profile. In bold *p* < 0.05.

**Table 6 cancers-12-02648-t006:** Akaike information criterion (AIC) and Bayesian information criterion (BIC) values of compared models.

Model	AIC	BIC
**TIP profile**	**68.3**	**74.5**
PD-L1 value	89.4	95.7
CD73 value	81.8	88.1
TILs value	70.0	76.1

Table 6 shows the evaluation of models based on TIP profile or single biomarkers using AIC and BIC. The best model is defined on the lower AIC and BIC value. AIC: Akaike information criterion; BIC: Bayesian information criterion; TIP: tissue immune profile. Bold represents the model with the lower AIC and BIC value.

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
