# Peer review of "Tissue Immune Profile: A Tool to Predict Response to Neoadjuvant Therapy in Triple Negative Breast Cancer"

_cancers, 2020, doi:10.3390/cancers12092648_

Round 1
Reviewer 1 Report
The current manuscript aimed to evaluate the association between the expression of some biomarkers in tissue samples and pathological complete response (pCR). Finally, they found that 3selected biomarkers as the predictive biomarker of pCR.
Major comments:
Introduction: add some more sentences to descript the association between pCR and survival outcome.
Results:
How do the authors define the threshold of different markers (sTILS ≥50%, PD-L1 ≥1% and CD73 ≤40%) expressed in tumor section?
In Table 1:
1. node-negative patients (the clinical N stage) is expressed as cN0 orcN-?
2. Move the following information to Table 2 " pCR: pathological complete response; RD: residual disease.".
Table 4:
replace " in bold p value <0.05. " by "Bold values denote statistical significance at the p < 0.05 level."
Table 5
remove "shows"
Table 6
replace " In bold the model with the lower AIC and BIC value. " by "Bold represents the model with the lower AIC and BIC value."
Author Response
R: Reviewer
A: Authors
Open Review
Comments and Suggestions for Authors
The current manuscript aimed to evaluate the association between the expression of some biomarkers in tissue samples and pathological complete response (pCR). Finally, they found that 3selected biomarkers as the predictive biomarker of pCR.
Major comments:
R: Introduction: add some more sentences to descript the association between pCR and survival outcome.
A: Thanks to the reviewer for the comment. We added a sentence and 3 references that extensively demonstrate the association between pCR and survival outcomes in different BC subtypes.
“Several meta-analysis confirmed the prognostic role of pCR, especially in aggressive BC subtypes such as triple-negative and HER2 positive.[8][9][10]” Line 62-63
R: Results: How do the authors define the threshold of different markers (sTILS ≥50%, PD-L1 ≥1% and CD73 ≤40%) expressed in tumor section?
Thanks to the reviewer for the comment. We defined the selected threshold of different markers as descripted in Materials and Methods of the original paper (section 4). In particular:
1) TILs: we defined H-TILs according to International TILs working group guidelines, who set the threshold between 50 and 60%: “Typically, the threshold of stromal lymphocytes for LPBC is around 50%–60% of the stromal surface area.”
As requested we added the sentence: “High TILs (H-TILs) were defined by the presence of TILs ≥ 50%, according to international guidelines.[34]” Lines 261-262;
2) PD-L1: we defined PD-L1 positivity according to recent international publications and guidelines on the use of immunotherapy in triple-negative breast cancer, which is the current used threshold in clinical practice. The sentence is: “The threshold of positivity for PD-L1 was set as the presence of immunostaining in ≥ 1% of IC as previously reported according to the recent approval of atezolizumab in metastatic TNBC.[36][37][38]”
3) CD73: actually, there is no international recognized and defined threshold for CD73 in breast cancer. In our recent study, we set the threshold to 40%, validated in our previous published paper, as described in Materials and Methods.
These 3 cut-offs for the selected biomarkers are associated with the presence of a favorable microenvironment and/or high response to treatment but represent different aspects of relation between cancer ad immune sistem. For this reason we combined them in a tissue immune-profile able to better predict pathological complete response to neoadjuvant chemotherapy.
R: In Table 1:
1. node-negative patients (the clinical N stage) is expressed as cN0 orcN-?
2. Move the following information to Table 2 " pCR: pathological complete response; RD: residual disease.".
A: 1. Thanks to the reviewer for the comment. The correct definition is cN0. Text and Tables have been updated.
2. corrected, thank you.
R: Table 4: replace " in bold p value <0.05. " by "Bold values denote statistical significance at the p < 0.05 level."
A: Thank to the reviewer for the suggestion. The text has been changed.
R: Table 5 remove "shows"
A: the text has been updated
R: Table 6 replace " In bold the model with the lower AIC and BIC value. " by "Bold represents the model with the lower AIC and BIC value."
A: Thanks again for the suggestion. The text has been corrected.
Reviewer 2 Report
The authors studied the tissue immune profile and correlated with pathologic complete response in patients with a triple negative breast cancer and treated with neoadjuvant chemotherapy. The patients were treated uniformly with a specific chemotherapy regimen. The immune profile composed (TIP) of tumor infiltrating lymphocytes, PD-L1 and CD73. They found that TIP independently correlated with pCR.
The methodology, slide from the small number of cases, is sound particularly with following the international guidelines for scoring tumor infiltrating lymphocytes in the use of the FDA approved clone for PDL-1. Moreover, all patients had similar neoadjuvant chemotherapy regimen.
I have the following comments:
There is a major issue with the study of having a small number of cases (n=70). As an evidence of this statement is that pCR did not correlate with the clinical T stage and with only borderline correlation with the clinical N stage
Although TIP+ is better than TILs value (table 6), the numbers are very close (68.3, 74.5 versus 70.0 and 76.1, respectively), making the value of this study limited.
The clinical TN stage should be combined into AJCC stage and correlate with the response to neoadjuvant chemotherapy
The type of the tumor (Triple negative breast cancer) should be mentioned the abstract
Author Response
R: Reviewer
A: Authors
Open Review
Comments and Suggestions for Authors
The authors studied the tissue immune profile and correlated with pathologic complete response in patients with a triple negative breast cancer and treated with neoadjuvant chemotherapy. The patients were treated uniformly with a specific chemotherapy regimen. The immune profile composed (TIP) of tumor infiltrating lymphocytes, PD-L1 and CD73. They found that TIP independently correlated with pCR.
The methodology, slide from the small number of cases, is sound particularly with following the international guidelines for scoring tumor infiltrating lymphocytes in the use of the FDA approved clone for PDL-1. Moreover, all patients had similar neoadjuvant chemotherapy regimen.
I have the following comments:
R: There is a major issue with the study of having a small number of cases (n=70). As an evidence of this statement is that pCR did not correlate with the clinical T stage and with only borderline correlation with the clinical N stage
A: We agree with the reviewer that the sample size is one the limit of the study. We have clearly emphasized this caveat in Discussion section.
line 217-224
“This study has some limitations. Firstly, it is a retrospective, single-institute trial. The relative small sample size of the population studied could be related to unexpected results, such as the borderline correlation that emerged between pCR and disease extension or the lack of association between PD-L1 and pCR. A prospective trial with a larger sample size is necessary to validate our results. Moreover, the benefit obtained using the profile over TILs alone (Table 6) is significant but limited, emphasizing the need for validation on a larger sample. Finally, further evaluation will be needed to confirm the predictive value of TIP after addition of immunotherapy to chemotherapy backbone.”
Thanks to the reviewer for the comment.
R: Although TIP+ is better than TILs value (table 6), the numbers are very close (68.3, 74.5 versus 70.0 and 76.1, respectively), making the value of this study limited.
A: Thanks to the reviewer. We added the following sentence to discuss this limit of our study.
Line 220-224
“A prospective trial with a larger sample size is necessary to validate our results. Moreover, the benefit obtained using the profile over TILs alone (Table 6) is significant but limited, emphasizing the need for validation on a larger sample. Finally, further evaluation will be needed to confirm the predictive value of TIP after addition of immunotherapy to chemotherapy backbone.”
R: The clinical TN stage should be combined into AJCC stage and correlate with the response to neoadjuvant chemotherapy
A: As requested by the reviewer, we combined TN stage into TNM stage and correlated it with response to neoadjuvant chemotherapy. We added the TNM stage description in Results (line 103-104 “Thirty-six (60%) patients had a clinical TNM II stage and 24 (40%) III stage.”; Table1 and Table 2) and Discussion (line 177-180: “Clinical TNM staging was not associated to pCR in our population. This result could be influenced by the relative small sample size, however is in line with several previous papers, highlighting the role of cancer biology over extension.[9][29][30]”)
R: The type of the tumor (Triple negative breast cancer) should be mentioned the abstract
A: Added
Reviewer 3 Report
The authors used TNBC surgical specimens to assess the rate of achievement of pathologic complete response (pCR) using TILs, PD-L1 expression, expression of CD73, and their combinations, TIPs, as indicators, and used Akaike information criterion and Bayesian information Criterion to show that TIPs can enrich predictions of pCR over each single indicator.
There seems to be no problems with the methods and results of the analysis, and it is clinically meaningful to improve the predictive ability of pCR for neoadjuvant chemotherapy.
On the other hand, a KEYNOTE-522 trial with statistically proven improvements in pCRs by adding pembrolizumab, an anti PD-1 antibody, to neoadjuvant chemotherapy was presented at ESMO2019 and published in NEJM this year, and the FDA is currently reviewing pembrolizumab for TNBC based on the results of this trial, and it is expected that immune-checkpoint inhibitory therapy will be incorporated into neoadjuvant therapy in the near future.
The relevance with this treatment (adding I-O) and potential usefulness of this study should be described in more detail later in Discussion part.
Reference
- Pembrolizumab for Early Triple-Negative Breast Cancer(N Engl J Med 2020; 382:810-821)
https://www.nejm.org/doi/full/10.1056/NEJMoa1910549 - Merck Announces Two US Regulatory Milestones for KEYTRUDA® (pembrolizumab) in Triple-Negative Breast Cancer (TNBC)
https://www.businesswire.com/news/home/20200730005275/en/Merck-Announces-Regulatory-Milestones-KEYTRUDA%C2%AE-pembrolizumab-Triple-Negative
Minor points
Page1 Line35: What does sTILs mean? need clarification.
Page8 Line247: PDL1 shoud be PD-L1
Author Response
R: Reviewer
A: Authors
Open Review
Comments and Suggestions for Authors
R: The authors used TNBC surgical specimens to assess the rate of achievement of pathologic complete response (pCR) using TILs, PD-L1 expression, expression of CD73, and their combinations, TIPs, as indicators, and used Akaike information criterion and Bayesian information Criterion to show that TIPs can enrich predictions of pCR over each single indicator.
There seems to be no problems with the methods and results of the analysis, and it is clinically meaningful to improve the predictive ability of pCR for neoadjuvant chemotherapy.
On the other hand, a KEYNOTE-522 trial with statistically proven improvements in pCRs by adding pembrolizumab, an anti PD-1 antibody, to neoadjuvant chemotherapy was presented at ESMO2019 and published in NEJM this year, and the FDA is currently reviewing pembrolizumab for TNBC based on the results of this trial, and it is expected that immune-checkpoint inhibitory therapy will be incorporated into neoadjuvant therapy in the near future.
The relevance with this treatment (adding I-O) and potential usefulness of this study should be described in more detail later in Discussion part.
Reference
Pembrolizumab for Early Triple-Negative Breast Cancer(N Engl J Med 2020; 382:810-821)
https://www.nejm.org/doi/full/10.1056/NEJMoa1910549
Merck Announces Two US Regulatory Milestones for KEYTRUDA® (pembrolizumab) in Triple-Negative Breast Cancer (TNBC)
https://www.businesswire.com/news/home/20200730005275/en/Merck-Announces-Regulatory-Milestones-KEYTRUDA%C2%AE-pembrolizumab-Triple-Negative
A: We thank the reviewer for the suggestion. We added the following sentences and references hoping they can meet the request.
“Recent clinical trials have shown promising activity of immune-checkpoint inhibitors on early TNBC, highlighting the need for new immune-related biomarkers. In Keynote 173 trial, the addition of pembrolizumab, an anti PD-1 agent, to standard chemotherapy produced a significant increase of pCR rate in an unselected population.[35] In the phase III GEPAR NUEVO trial, the addition of durvalumab, an another anti PD-1 agent, resulted in an increased rate of pCR from 44% to 53%.[36] In contrast, preliminary results from NeoTRIP trial show no significant difference in pCR adding atezolizumab to a backbone of chemotherapy with weekly carboplatin plus nab-paclitaxel in the same setting.[37] Finally, the results from phase III Keynote-522 trial show that adding pembrolizumab to standard neoadjuvant chemotherapy increased the pCR rate and improve DFS, regardless PD-L1 expression.[38] Based on these results, is expected that anti PD-1 agents will soon be introduced in neoadjuvant treatment of early TNBC. As a future perspective, an immune-profile, such as TIP, could improve the selection of patients who could benefit from combination strategies with anti PD-1 agents, considering the high cost and additional toxicities related to therapy escalation.”
Minor points
R: Page1 Line35: What does sTILs mean? need clarification.
A: Thanks to the reviewer for the comment. We modified the text with TILs as it was in the rest of the paper.
R: Page8 Line247: PDL1 shoud be PD-L1
A: we apologise for the error and thanks the reviewer for the suggestion. The text has been updated.
Round 2
Reviewer 1 Report
The paper is accepted without any further changes.
Reviewer 2 Report
none